# Virulence of *Metarhizium rileyi* Is Determined by Its Growth and Antioxidant Stress and the Protective and Detoxifying Enzymes of *Spodoptera frugiperda*

**DOI:** 10.3390/insects14030260

**Published:** 2023-03-06

**Authors:** Jixin Pang, Yuejin Peng, Teng Di, Guangzu Du, Bin Chen

**Affiliations:** Yunnan State Key Laboratory of Conservation and Utilization of Biological Resources, College of Plant Protection, Yunnan Agricultural University, Kunming 650201, China

**Keywords:** antioxidant stress, detoxifying enzyme, growth, *Metarhizium rileyi*, protective enzyme, *Spodoptera frugiperda*

## Abstract

**Simple Summary:**

*Metarhizium rileyi* (XSBN200920 and HNQLZ200714) were isolated from diseased *S. frugiperda*. This study not only elucidated the pathogenic process of *Metahizium rileyi* from the activities of the host detoxifying enzyme and protective enzyme but also identified the key factors responsible for the variation in virulence between XSBN200920 and HNQLZ200714 based on the growth of pathogenic fungi, the expression of antioxidant stress genes (SOD and CAT genes), and the expression of antioxidant enzymes. the high virulence of *M. rileyi* XSBN200920 was not only determined by the expression levels of protective and detoxifying enzymes of the host but also regulated by the growth of entomogenic fungi and the resistance to the oxidative stress against *S. frugiperda* at different developmental stages. This study provides a theoretical basis for the systematic control of *Spodoptera frugiperda* using *Metarhizium rileyi*.

**Abstract:**

*Spodoptera frugiperda* is one of the most destructive crop pests in the world. *Metarhizium rileyi* is an entomopathogenic fungus specific for noctuid pests and is a very promising prospect in biological control against *S. frugiperda*. Two *M. rileyi* strains (XSBN200920 and HNQLZ200714) isolated from infected *S. frugiperda* were used to evaluate the virulence and biocontrol potential to different stages and instars of *S. frugiperda*. The results showed that XSBN200920 was significantly more virulent than HNQLZ200714 to eggs, larvae, pupae, and adults of *S. frugiperda*. In the larvae infected with the two *M. rileyi* strains, the activity of three protective enzymes (including peroxidase (POD), superoxide dismutase (SOD), catalase (CAT)) and two detoxifying enzymes (including glutathione-S transferase (GST) and carboxylesterase (CarE)) increased firstly and then decreased. The expression levels of protective enzymes and detoxification enzymes in larvae treated with XSBN200920 were greater than with HNQLZ200714. Furthermore, antioxidant stress-related gene (*MrSOD* and *MrCAT* family genes) expression in the two strains was measured by RT-qPCR (real-time quantitative PCR). The expression of these genes was significantly higher in the XSBN200920 strain compared to HNQLZ200714. There were also significant differences in the sensitivity of the two strains to the growth of different carbon and nitrogen sources and oxidative stress agents. In addition, the activity expression of antioxidant enzymes on the third day of culturing in XSBN200920 was significantly higher than with HNQLZ200714. In summary, the high virulence of *M. rileyi* XSBN200920 was not only determined by the expression levels of protective and detoxifying enzymes of the host but also regulated by the growth of entomogenic fungi and the resistance to the oxidative stress against *S. frugiperda* at different stages and instars. This study provides a theoretical fundament for the systematic control of *Spodoptera frugiperda* using *Metarhizium rileyi*.

## 1. Introduction

*Spodoptera frugiperda* (J. E. Smith) (Lepidoptera: Noctuidae) is a devastating migratory pest that can be found throughout the world [1]. Since it has many host species, a wide range of distribution areas, and strong reproductive ability, it not only causes huge economic losses in North and South American corn-growing areas but also causes serious harm to parts of Africa and Asia. Therefore, *S. frugiperda* poses a threat to food security and sustainable crop productivity [2,3,4,5]. However, the long-term and large-scale use of chemical insecticides to mediate this pest has not been effective and resulted in high resistance to many traditional insecticides [6]. In addition, the long-term use of chemical pesticides is likely to cause further environmental pollution, destroy the ecological balance, and harm non-target organisms. Therefore, chemical control is not an environmentally friendly control strategy for *S. frugiperda* will face significant challenges in controlling *S. frugiperda* populations [7]. Seeking new control methods would be the most effective strategy to reduce *S. frugiperda* damage [8].

Entomopathogenic fungi are regarded as the primary source of environmentally friendly fungicides worldwide since they are the major biological factors that regulate arthropod populations [9]. *Metarhizium rileyi* (synonymous with *Nomuraea rileyi*) (Farl.) Samson [10] is a filamentous fungus that is indispensable in the field control of Lepidoptera pests throughout the world [11]. The infection with *M. rileyi* involves the general recognition of the host, attachment of conidia to the host body surface, the formation of appressorium and germs, penetration of bud tubes through the host epidermis, and colonization of the parasite in the haemocoele [12]. This process is influenced by a variety of factors, ultimately leading to the insect host death [13,14]. Following the invasion of the host body wall, the entomological fungi enter the blood cavity and also regulate their superoxide dismutase (SOD) and catalase (CAT) in response to the environmental stress [5,15,16]. Concomitantly, the fungi attack the protective enzyme (superoxide dismutase (SOD), peroxidase (POD), catalase (CAT) enzymes) and detoxification enzyme (glutathione-S transferase (GST), carboxylesterase (CarE) enzymes) defense systems in the insect body. Therefore, the insect cannot maintain a dynamic equilibrium state, and the fungus destroys a series of defense functions in the host body, obtains nutrients from the host body for reproduction, and eventually kills the host [17].

In this study, to understand the underlying reasons for the virulence differences between different Entomopathogenic fungi, two *Metarhizium rileyi* strains were isolated and identified from *S. frugiperda* in the affected areas in Yunnan Province, China. *Metarhizium rileyi* was detected in the four stages of development of *S. frugiperda* following metamorphosis. The host protective enzyme and detoxifying enzyme systems as well as the function of the antioxidant enzyme system of pathogenic fungi were investigated to identify the underlying variations in virulence of different *M. rileyi strains*. This study, therefore, provides a theoretical fundament to control *Spodoptera frugiperda* via biotechnology.

## 2. Materials and Methods

### 2.1. The Fungi and Insects

*Metarhizium rileyi* strains were collected from infected larval *S. frugiperda* on maize in the main maize growing areas of Xishuangbanna Dai Autonomous Prefecture (21°08′ N, 99°56′ E) and Qinglong Town (24°13′ N, 103°06′ E) of Yuxi City, Yunnan Province, China in 2019. *S. frugiperda* was collected from the maize affected area in Mile County, Yuxi City, Yunnan Province, China. Insects used in this experiment were all reared and bred on corn leaves in this laboratory for more than 10 generations of eggs, third, and fifth instar larvae, pupae Adults (male and female) of *S. frugiperda were determined by virulence of M. rileyi*.

### 2.2. Fungal Isolation and Culture

Following the complete sporulation of the *M. rileyi* from the collected insects, spores (10^7^ conidia/mL) were gently collected with an inoculation needle on an ultra-clean workbench onto potato dextrose agar (PDA) solid medium and cultured in a constant temperature incubator with a relative humidity (RH) of 70% and 16 h light/8 h dark photoperiod. After 5 days of mycelia growth, a single colony was selected and cultured on a clean PDA medium using the striating method [18]. After 10 days, the spores were stored at −80 °C degrees for future use. Sabouraud dextrose agar plus yeast extract (SDAY) medium (1% yeast extract, 4% glucose, 1% peptone, 1.5% agar) and PDA medium (200 g/L potato, 2% glucose, 1.5% agar) were used for fungal isolation and culture. Czapek-Dox medium (CZA: 3% sucrose, 0.1% K_2_HPO_4_, 0.05% KCl, 0.3% NaNO_3_, 0.05% MgSO_4_, 0.001% FeSO_4_, 1.5% agar) was used as a chemical-defined medium. The two strains XSBN200920 and HNQLZ200714 were identified to be deposited in our laboratory.

### 2.3. Molecular Identification and Phylogenetic Analysis of Metarhizium rileyi

The genomic DNA was extracted by the CTAB method [19]. The fungal universal primers ITS1 (5’-TCCGTAGGTGAACCTGCGG-3’) and ITS4 (5’-TCCTCCGCTTATTGATATGC-3’) were used for PCR amplification of the rDNA-ITS strain. The PCR products were sent to Sangon Bioengineering Co., Ltd. (Shanghai, China) for sequencing. The sequenced ITS sequence was submitted to the NCBI website (https://blast.ncbi.nlm.nih.gov/Blast.cgi, accessed on 20 May 2020). The species and sequence items with high homology were retrieved by BLAST function. The Clustalx function of MEGA7.0 software was used to compare the sequences, and the items with high homology were screened out. The phylogenetic tree was constructed by neighbor-joining [20].

### 2.4. Morphological Observation and Colony Growth Rate Measurement of Metarhizium rileyi

Spore suspensions with a concentration of 1.0 × 10^7^ conidia/mL were prepared, and 1 µL of conidia from the XSBN200920 or HNQLZ200714 strains was inoculated to the center of SDAY and PDA medium by the spot joint method, respectively, and incubated in a light incubator at 25 °C for 7 days at 16 h light/8 h dark photoperiod. Colony color, morphology, texture, and edge shape were photographed, and colony diameter was measured [21]. Conidia of XSBN200920 and HNQLZ200714 strains were diluted to 1.0 × 10^6^ conidia/mL. A total of 5 µL of the spore suspension was removed for culture on different phenotypic assay plates for 7 days. Colony diameters of XSBN200920 and HNQLZ200714 strains were measured, and photographs were taken according to the experimental requirements of each group. The experiment was repeated in triplicate with three parallel controls each time. In this study, the effects of different carbon sources, nitrogen sources, and oxidative stress agents on fungal growth were compared using the CZA medium to measure the colony growth rate. The carbon sources detected included 3.0% glucose, 3.0% sucrose, 3.0% lactose, and 3.0% maltose, respectively. The nitrogen sources included 0.3% NH_4_Cl, 0.3% gelatin, 0.3% peptone, and 0.3% NaNO_3_, respectively. The stress agents included 4 mM H_2_O_2_ and 0.03 mM menadione, respectively. Relative growth inhibition (RGI) of each strain to each stressor was estimated as RGI (%) = (Dc − Dt)/Dc × 100, where Dc and Dt were the diameters of the control and stressed colonies, respectively.

### 2.5. Determination of Strain Virulence to Different Stages of *Spodoptera frugiperda*

Virulence of the two strains at different concentrations to the eggs of *S. frugiperda*. The *S. frugiperda* eggs were inoculated on maize leaves. Eggs from the same day were selected, 50 eggs were retained in each egg block, and the excess eggs were removed. Conidia of XSBN200920 and HNQLZ200714 strains were inoculated using the spray method (1.0 × 10^8^ conidia/mL). Three replicates were performed for each treatment, and a 0.02% Tween-80 treatment group was used as the blank control. The treated egg blocks were placed in disposable Petri dishes containing fresh corn leaves. These dishes were continuously observed to record the hatching of the eggs and the survival of the newly hatched larvae. The egg-hatching rate and the mortality rate of the newly hatched larvae were calculated.

Virulence of the two strains to larva of *S. frugiperda*. Conidia of the two strains were prepared into 10^4^, 10^5^, 10^6^, 10^7^, and 10^8^ conidia/mL concentrations and 0.02% Tween-80 solution was used as the control. Healthy third and fifth instar larvae of *S. frugiperda* of the same size were selected and added into the prepared spore suspension, soaked for 15 s, and then removed. The excess water on the surface of the larvae was absorbed with sterile filter paper. A single head was put into a 12-well culture plate following disinfection with 75% alcohol. Fresh corn leaves were added to the hole where the larvae were placed. The fresh corn leaves were replaced daily. The control was treated with 0.05% aseptic water solution with a soil temperature of −80 °C. This experiment was conducted in triplicate for each treatment of 35 specimens. Following treatment, the larvae were fed in an artificial climate box at 25 °C, 75% RH and a photoperiod of 16 h light/8 h dark. The number of dead insects was recorded daily, and the dead insects were kept in a petri dish for moisture observation to confirm whether the test strain was infected and killed. The cumulative death rate of insects = number of dead insects/total number of insects × 100.

Virulence of two strains to pupae of *S. frugiperda*. One-day-old pupae of a similar size were selected and added to the prepared spore suspension by the cuticle infection. In the cuticle infection assay, the pupae were inoculated by immersing in the conidial suspension for 15 s. After 15 s of impregnation, the pupae were removed, and the excess water on the surface of the worm was dried with sterile filter paper. The single head was placed in a sterilized petri dish (9 cm in diameter) with wet filter paper at the bottom of the dish [22]. This experiment was conducted in triplicate for each of the 30 treated pupae. The number of dead pupae and the emergence of pupae were recorded daily.

Virulence of two strains at different concentrations to adult of *S. frugiperda*. Female and male adults of two-day-old armyworm were selected and inoculated with 10^6^, 10^7^, and 10^8^ conidia/mL conidia of the above two strains by spraying in the conidial suspension for 10 s. Following inoculation, the female and male adults were placed in a plastic film cage (20 × 20 × 20 cm). The adults were reared in a light incubator at 25 °C, 70% RH and a 16 h light/8 h dark photoperiod. This experiment for female and male adults was conducted in triplicate, with 30 heads per replicate treatment. The number of dead adults was recorded from the 3rd to the 8th day, and the dead insects were cultured with moisture in time, and the morphology of infected adults was recorded by taking photos.

### 2.6. Determining Activity of the Main Protective and Detoxifying Enzymes in S. frugiperda

The 3rd instar larvae were inoculated with a spore suspension with a concentration of 10^8^ conidia/mL. The third instar larvae were treated for 24, 36, 48, and 60 h. A sterilized 0.02% Tween-80 solution was used as the control. There were 30 worms per replicate, and 3 replicates were taken from each treatment. The insects were rinsed with normal saline and dried with sterile filter paper, then transferred to a sterile pre-cooled mortar and ground in an ice bath. Then, 1 mL normal saline was added to form homogenates and transferred to a pre-cooled 2 mL centrifuge tube and centrifuged at 4 °C at 5000 rpm for 8 min, and the supernatant was carefully absorbed with a pipetting gun for use. Protective enzymes (SOD, CAT, POD), detoxification enzymes (CarE, GST), and total protein concentration were determined using the detection kit of Jianguo Bioengineering Institute (Nanjing, China), and the enzyme activities of the above enzymes were determined according to the instructions [23].

### 2.7. Determining SOD and CAT Activity in Fungi

*Metarhizium rileyi* were grown for three days under the SDAY at 25 °C. Each PDA plate was covered with cellophane, and 10^7^ conidia/mL suspension was added in 100 μL equal parts for culture initiation. Protein extracts (mean 0.5 g) were isolated from mycelium cultures and quantified using the bicinchoninic acid (BCA) protein detection kit (KeyGen Biotech, Nanjing, China). The CAT and SOD assay kits (Sigma-Aldrich, St. Louis, MO, USA) were used to determine total enzymatic activity (U/mg protein extract) in three protein samples of each strain [24].

### 2.8. Fungal RNA Extraction and RT-qPCR

The XSBN200920 and HNQLZ200714 strains were prepared into spore suspensions with a concentration of 10^7^ spores/mL. A total of 100 μL was absorbed using a pipetting gun, added to a cellophane-covered SMAY plate, and incubated at 25 °C. After three days, mycelia (about 0.5 g) were selected from each strain, and total RNA was extracted from stressed or non-stressed cells using the RNAiso™ Plus reagent (TaKaRa, Dalian, China) and reverse-transcribed into cDNA using PrimeScript^®^RT reagent kit (TaKaRa, Dalian, China) [16]. RT primers were designed using the NCBI website, and each cDNA sample was diluted 20 times as a template for real-time quantitative PCR (qRT-PCR) analysis. The primers used for qRT-PCR are presented in Table 1.

### 2.9. Data Analysis

The observations and measurements of percentage of bioassays about insect eggs, larvae, pupae, and adults, protective enzymes and detoxification enzymes, and methodologies of fungal strains in three repeated experiments were analyzed using one-way and two-factor analysis of variance (ANOVA), followed by Tukey’s honest significance difference test (Tukey’s HSD) to compare the mean values between the fungal strains. The analyses were conducted using Graphpad Prism 8.4.2 at the different level of significance.

## 3. Results

### 3.1. XSBN200920 and HNQLZ200714 Isolated from the Affected *Spodoptera frugiperda* Were Identified as Metarhizium rileyi

The strains (XSBN200920 and HNQLZ200714) were isolated from diseased *S. frugiperda* (Figure 1A). Phylogenetic analysis based on ITS sequence alignment showed that the two strains were closely related to and identified as *M. rileyi* (Figure 1B). The colony morphology results showed that the colony growth rate of XSBN200920 was significantly faster than that of HNQLZ200714 (Figure 1C). On the seventh day of culture on SDAY medium, the colony diameter of XSBN200920 was 2.0 ± 0.15 cm (mean ± SD) and approximately 11% larger than HNQLZ200714 which was 1.78 ± 0.04 cm. On the PDA medium, the colony diameter of XSBN200920 was 1.78 ± 0.04 cm and significantly larger (by 28%) compared to HNQLZ200714 (1.29 ± 0.05 cm).

### 3.2. Virulence of Metarhizium rileyi XSBN200920 and HNQLZ200714 for Different Stages of Spodoptera frugiperda

The fungal virulence of eggs and newly hatched larvae were assessed using egg-hatching and post-hatching larval survival rates. The egg masses of *S. frugiperda are presented in* Figure 2A (a and b). The *M. rileyi* XSBN200920 and HNQLZ200714 strains exhibited significant differences in egg hatchability and neonatal larval mortality of *S. frugiperda* (Figure 2B). Compared with the control group, the hatching rate of *S. frugiperda and* eggs treated with XSBN200920 decreased by 32%, but that of HNQLZ200714 decreased by 18% (χ^2^ = 6.725, *df* = 2, *p* < 0.05). Significant differences were detected in the mortality of the newly hatched larvae of the two strains at 1.0 × 10^4^ spores/mL (Figure 2C). In contrast to the HNQLZ200714, the mortality rates of emerging larvae of XSBN200920 strain in groups 10^4^, 10^5^, 10^6^, 10^7^, and 10^8^ increased by 8.9, 10.1, 21.7, 13.8, and 14.7%, respectively (χ^2^ = 73.27, *df* = 4, *p* < 0.05).

The cumulative mortality rate of XSBN200920 and the third and fifth instars of *M. rileyi* against *S. frugiperda* larvae was significantly higher than that of HNQLZ200714 in the virulence assessment. The infected larva of *S. frugiperda are shown in*
Figure 2A (c and d). The results of the virulence of the two strains against the third instar larvae demonstrated that the cumulative mortality of the third instar larvae in groups 10^4^, 10^5^, 10^6^, 10^7^, and 10^8^ of the XSBN200920 treatment group was higher than in the HNQLZ200714 (Figure 2D,E). Statistical analysis was performed to obtain the relevant parameters of virulence of the two *M. rileyi* strains against the third instar larvae of *S. frugiperda* (Table 2).

Similarly, the virulence results of the two strains against the fifth instar larvae demonstrated that the cumulative mortality in groups 10^4^, 10^5^, 10^6^, 10^7^, and 10^8^ of the XSBN200920 treatment group was higher than in HNQLZ200714 (Figure 2F,G). On the seventh day following treatment with 1.0 × 10^7^ or 1.0 × 10^8^ spores/mL, the cumulative mortality rate of the fifth instar larvae was 53.5 and 64%, respectively. The cumulative mortality rate of the HNQLZ200714 strains was 57.6% when the concentration was 1.0 × 10^8^ spores/mL. Statistical analysis was performed to obtain the relevant parameters of virulence of the two *M. rileyi strains against* the fifth instar larvae of *S. frugiperda* (Table 3).

The results demonstrated that XSBN200920 and HNQLZ200714 strains displayed significant effects on the *S. frugiperda* pupa emergence rate. The effects of the XSBN200920 strain on the pupa emergence rate were significantly higher than those of the HNQLZ200714 strain. When the 1.0 × 10^8^ spores/mL concentration was used, the emergence rates of pupae in the XSBN200920 and HNQLZ200714 treatment groups were 59.3 ± 1.16 and 70.8 ± 1.33%, respectively (Figure 2H). Figure 2A (e and f) shows the pupae of a newly treated *S. frugiperda* and the pupae of a dead insect five days following the treatment.

The XSBN200920 strain displayed greater virulence, while the HNQLZ200714 strain displayed lower virulence in female and male adults of *S. frugiperda* (Figure 2I,J). Figure 2A (g and h) shows the state of an infected adult two and five days following death. The mortality rate of female adults inoculated with XSBN200920 at 10^6^, 10^7^, and 10^8^ was 23.6, 22.3, and 47.2% higher than with HNQLZ200714 (χ^2^ = 11.91, *df* = 2, *p* < 0.01). Similarly, the mortality rate of male adults inoculated with XSBN200920 at 10^6^, 10^7^, and 10^8^ was 16.7, 31.4, and 43.3% higher than with HNQLZ200714 (χ^2^ = 11.84, *df* =2, *p* < 0.01).

### 3.3. The Protective Enzyme Activity of S. *frugiperda* Larvae *Infected with* XSBN200920 and HNQLZ200714 Was Unbalanced

The activity of the three protective enzymes in the larvae of *S. frugiperda* first increased and then decreased following *M. rileyi infection*. However, the effects of XSBN200920 and HNQLZ200714 on the protective enzyme activity of *S. frugiperda* larvae varied. According to the changes in POD enzyme activity (Figure 3A), POD activity in XSBN200920 and HNQLZ200714 strains treated with insects was significantly different (*F* = 0.7136, *df* = 2, *p* < 0.05) compared with the control group. Within 36 h of treatment, the POD activity in the XSBN200920 treatment group was higher than in HNQLZ200714 following *S. frugiperda treatment*. The POD enzyme activity of strain HNQLZ200714 exceeded that of XSBN200920 by 34.0 ±1 and 32.0 ±1 U/mg, respectively, at the 48th hour following inoculation. At 60 h, there was no significant difference in POD activity between the two experimental strains. The SOD activity (Figure 3B) in XSBN200920 and HNQLZ200714 strains treated with *S. frugiperda* varied significantly (*F* = 4.143, *df* = 2, *p* < 0.05) compared to the control group. The maximum values were 26.8 ± 0.29 and 28.3 ± 0.58 U/mg at 36 h, respectively. At 60 h, the enzyme activity in HNQLZ200714 exceeded XSBN200920. Significant differences were observed in CAT activity (Figure 3C) between XSBN200920 and HNQLZ200714 strains treated with *S. frugiperda* (*F* = 7.804, *df* = 2, *p* < 0.05) compared to the control group. The CAT activity in XSBN200920 and HNQLZ200714 strains increased first and then decreased, reaching the maximum value of 35.8 ± 0.76 U/g Hb at 36 h after infection by XSBN200920. The maximum value in HNQLZ200714 was 36.3 ± 1.53 U/g Hb at the 48 h following infection.

### 3.4. XSBN200920 and HNQLZ200714 Demonstrated Significant Effects on the Detoxifying Enzyme Activity of *S*. *frugiperda* Larvae

The changes in CarE activity in the host after treatment with *M. rileyi* are presented in Figure 3D. At 36 h following treatment with XSBN200920 or HNQLZ200714, the maximum CarE activity in the host reached 2.37 ± 0.35 and 2.72 ± 0.03 U/mg, respectively, and began to decline. At about 48 h, the enzyme activity of the treatment group was lower compared to the control group and showed a significant difference at 60 h (*F* = 249.5, *df* = 2, *p* < 0.01).

Compared with the control group, GST activity in the treatment group varied significantly at different periods (*F* = 10.05, *df* = 2, *p* < 0.01) (Figure 3E). The difference between the two treatments was not obvious before 48 h. At 48 h, XSBN200920 and HNQLZ200714 reached the maximum value, which was 2.04 and 2.10 times the control group, respectively. At 60 h, GST activity in the XSBN200920 treatment group (64.0 ± 1 U/mg) was slightly higher than in HNQLZ200714 (57.0 ± 1 U/mg).

### 3.5. XSBN200920 and HNQLZ200714 Showed Significant Differences in Response to Carbon Sources, Nitrogen Sources, and Oxidative Stress Agents

XSBN200920 and HNQLZ200714 showed significant growth differences in response to various carbon sources (*F* = 87.56, *df* = 2, *p* < 0.05) (Figure 4A). On the seventh day, the colony diameters of XSBN200920 on the sucrose, lactose, maltose, and glucose plates as the sole carbon source were 3.83, 1.13, 1.32, and 2.72 times that of HNQLZ200714, respectively. Similarly, the two strains displayed significant differences in growth in response to various nitrogen sources (*F* = 10.69, *df* = 2, *p* < 0.01) (Figure 4B). The colony diameters of XSBN200920 on peptone, NH_4_Cl, gelatin, and NaNO_3_ as the sole nitrogen source were 2.0, 2.59, 1.91, and 3.82 times that of HNQLZ200714. In terms of antioxidant stress, the RGI rate of the XSBN200920 strain was significantly lower than that of the HNQLZ200714 strain on menadione (χ^2^ = 62.83, *df* = 2, *p* < 0.001) and H_2_O_2_ (χ^2^ = 18.27, *df* = 2, *p* < 0.01) plates (Figure 4C).

### 3.6. XSBN200920 and HNQLZ200714 Showed Significant Differences in MrSOD and MrCAT Family Gene Expression and Antioxidant Enzyme Activity

The RT-qPCR results demonstrated that on the third day of culture, *MrCAT1*, *MrCAT2*, and *MrCAT3* expression in XSBN200920 was 39.6, 55, and 37 times higher, respectively, than in HNQLZ200714 (Figure 4D). The *MrSOD* (*F* = 2316, *df* = 1, *p* < 0.001) and *MrCAT* gene expression (*F* = 57160, *df* = 1, *p* < 0.001) in the XSBN200920 strain was significantly higher than in HNQLZ200714. *MrSOD1*, *MrSOD2*, *MrSOD3*, and *MrSOD4* expression in XSBN200920 was 18.7, 2.68, 76.2, and 8.68 times higher, respectively, than in HNQLZ200714 (Figure 4E). Fungal enzyme activity in XSBN200920 and HNQLZ200714 was significantly different (*F* = 51.39, *df* = 1, *p* < 0.05). The SOD and CAT activity in the XSBN200920 strain was 1.67 and 1.91 times higher than in HNQLZ200714, respectively (Figure 4F).

## 4. Discussion

In this work, XSBN200920 and HNQLZ200714 strains isolated from mummified *S. frugiperda* were used to measure and evaluate the virulence of egg hatching rate, larval mortality rate, eclosion rate, and adult mortality rate of *S. frugiperda* in various periods. The results showed that the toxicity of XSBN200920 to different stages of *S. frugiperda* was significantly higher than that of HNQLZ200714. The virulence of different strains to the same host insect often vary greatly [25]. The results of this study were consistent with those previously reported [26]. There were significant differences in the mortality rate of *M. rileyi* strains against *S. frugiperda*. This may be due to the differences in the body surface structure and internal defense mechanisms among different insect stages. In addition, the structure or composition of the integument at different developmental stages affects the infection and penetration of *M. rileyi*. On the other hand, the variation in the ability and number of surface-attached conidia of the strains against different developmental stages of insects may result in significant differences in the virulence of the two *M. rileyi* strains against different developmental stages of *S. frugiperda* [27].The data of the virulence of *M. rileyi* to different stages of *S. frugiperda* showed that the higher the concentration of spore suspension, the higher the mortality rate of the newly hatched larvae, the third and fifth instar larvae, and the adults, and pupation was significantly increased. This result is consistent with previous reports [28]. However, there are significant differences in the pathogenicity of XSBN200920 and HNQLZ200714 strains to various stages of *S. frugiperda*. It indicated that the two strains showed a certain degree of stability in virulence to the host.

The activity of POD, SOD, and CAT enzymes in *S. frugiperda inoculated* with two fungi demonstrated first an increase and then a decrease trend (Figure 3A–C). These changes may be due to the activation of the protective enzyme systems of the host following the infection by pathogenic fungi to help the host survive the stress period. Similarly, the tendency of CarE and GST activity in the insects increased and then decreased following the treatment with *M. rileyi* (Figure 3D,E) which may be an immune defense mechanism of the host. However, CarE and GST activity declined 36 and 48 h following treatment, respectively. As previously reported [29,30], these results suggest that insect detoxifying enzyme systems cannot clear the toxins produced by pathogenic fungi from the body.

SOD is a metal protein which can neutralize active superoxide (ROS) in the cell, which is one of the key enzymes in the cell defense system to fight against superoxide damage [31,32]. In *Beauveria bassiana* (Bals.) Vuill., *SOD* and *CAT* genes play an essential role in the host infection process of the fungus. In addition, they are vital for the antioxidant and ultraviolet tolerance of cells [32,33]. Studies have shown that *SOD* and *CAT* genes have important effects on the growth and development, tolerance, and virulence of fungi [34]. When entomopathogenic fungi invade the haemocoele of insects, they also trigger the expression of a series of proteins that aid in colonizing the organism [35]. The current study demonstrated that *SOD* and *CAT* relative expression in the XSBN200920 strain was significantly higher than in HNQLZ200714. These were consistent with the results of fungal resistance to menadione and H_2_O_2_ oxidative stress. The potent antioxidant capacity of *M. rileyi* XSBN200920 plays a significant role in the killing process of *S. frugiperda*. In addition, the growth and development of pathogenic fungi are also influential factors affecting the virulence potential of fungi. In *Beauveria bassiana*, the deletion of genes associated with growth and development directly leads to a severe loss of fungal virulence [36,37]. Based on the results of this study, the high virulence of the XSBN200920 strain is clearly a result of its robust growth characteristics.

The results of our work showed that *Metarhizium rileyi* XSBN200920 and HNQLZ200714 showed significant differences in their pathogenicity against eggs, larvae, pupae, and adults of *Spodoptera frugiperda*. The two fungi affected the expression of protective enzymes and detoxification enzymes in the larvae of *Spodoptera frugiperda* to different degrees, resulting in differences of virulence. Moreover, the differences in the growth and antioxidant stress of XSBN200920 and HNQLZ200714 strains were consistent with the results of the expression levels of genes related to the regulation of growth and development of the fungi. Compared to previously published studies [33], this work is characterized by the first systematic evaluation of the virulence of two *Metarhizium rileyi strains against* all developmental stages of *Spodoptera frugiperda*, including eggs, larvae, pupae, and adults. More importantly, this study not only elucidated the pathogenic process of *Metahizium rileyi from* the activity of the host detoxifying enzyme and protective enzyme but also identified the key factors responsible for the variation in virulence between XSBN200920 and HNQLZ200714 based on the growth of pathogenic fungi, the expression of antioxidant stress genes (*SOD* and *CAT* genes), and the expression of antioxidant enzymes. The results of this study will help in gaining an in-depth and comprehensive understanding of the underlying mechanisms of the infection and death of entomopathogenic fungi.

## Figures and Tables

**Figure 1 insects-14-00260-f001:**
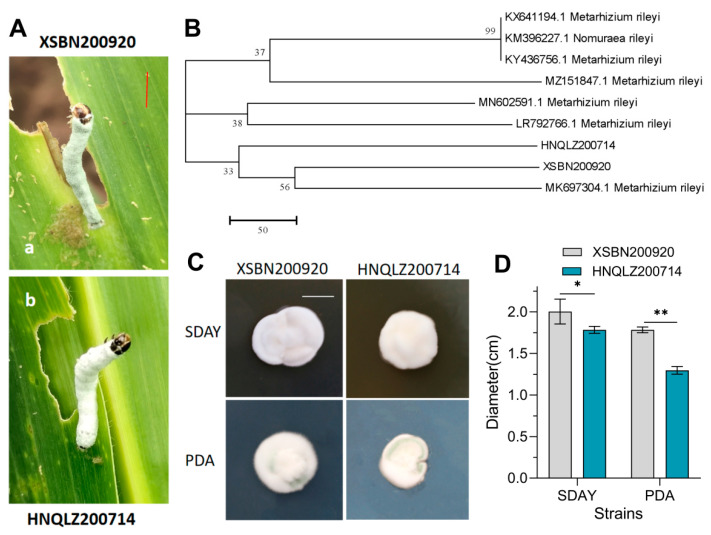
Isolation and identification of *Metarhizium rileyi* XSBN200920 and HNQLZ200214: (**A**) diseased *Spodoptera frugiperda* found on maize plants; a and b are *Spodoptera frugiperda* (Bar: 1 cm) collected from Xishuangbanna (XSBN200920H) and Yuxi City (HNQLZ200214), respectively; (**B**) phylogenetic analysis of strains XSBN200920 and HNQLZ200214; relationships are depicted from neighbor joining analysis, and the bootstrap values > 50% from 1000 replicates are shown at the supported branch; (**C**,**D**) are colony morphology and diameters of XSBN200920 and HNQLZ200214 strains on SDAY and PDA plates, respectively; Tukey’s honestly significant difference [HSD]: *p* < 0.05 (*), *p* < 0.01 (**); error bars: standard deviation.

**Figure 2 insects-14-00260-f002:**
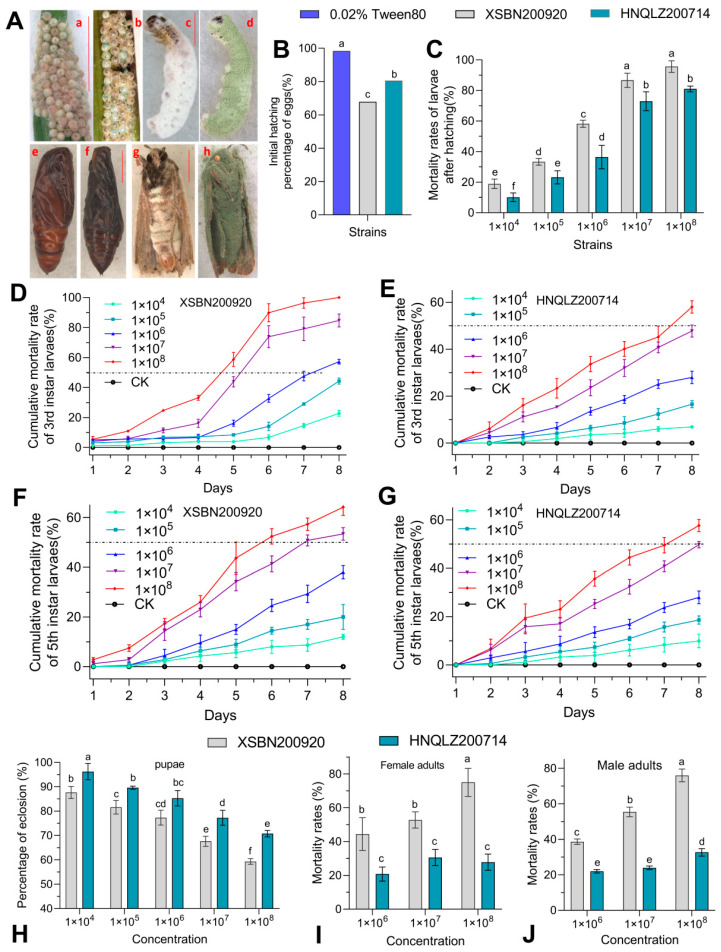
Assessment of the virulence of *Metarhizium rileyi* XSBN200920 and HNQLZ200214 against various stages and instars of *Spodoptera frugiperda*: (**A**) various states and instars of *S. frugiperda*; a and b are eggs treated with 0.02% tween80 and spore suspensions (10^8^ conidia/mL) of *M. rileyi* after 2 days, respectively; c and d are larvae of the 3rd instar inoculated with *M. rileyi* on days 2 and 5, respectively; e and f are pupae treated with 0.02% tween80 and spore suspensions (10^8^ condia/mL) of *M. riley* for 5 days, respectively; g and h are the adults infected with *M. rileyi* (Bar: 1 cm) on day 2 and day 5, respectively; (**B**,**C**) are thepost-egg hatching rate and mortality rate of hatched larvae infected by XSBN200920 and HNQLZ200214 strains of *S. frugiperda*, respectively; (**D**–**G**) are the cumulative mortality rates of the third and fifth instars of *S. frugiperda* infected with conidia of XSBN200920 and HNQLZ200214 at different concentrations by body wall infection, respectively; (**H**) emergence rate of pupae inoculated with different concentrations of fungal spores; (**I**,**J**) are the mortality rates of male and female *S. frugiperda* after fungal spores of different concentrations infected them. Tukey’s honestly significant difference [HSD]: *p* < 0.05; error bars: standard deviation.

**Figure 3 insects-14-00260-f003:**
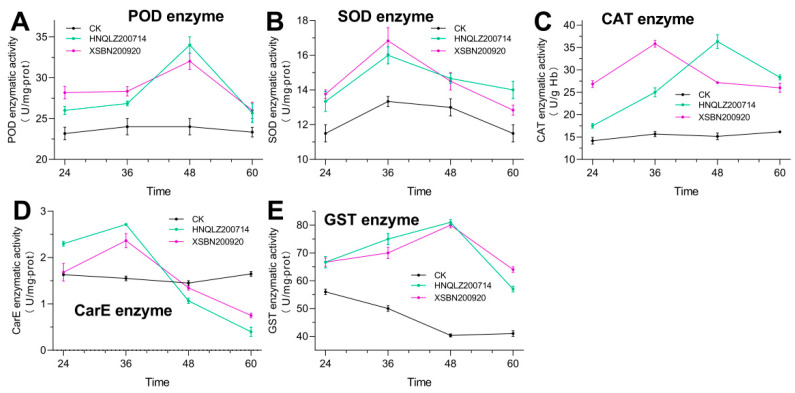
Protective enzyme and detoxifying enzyme activities of *Spodoptera frugiperda* larvae infected with *Metarhizium rileyi*: (**A**–**E**) are the activities of POD enzyme, SOD enzyme, CAT enzyme, CarE enzyme, and GST enzyme in the host from 24 h to 60 h after infection by XSBN200920 and HNQLZ200214 strains, respectively; error bars: standard deviation.

**Figure 4 insects-14-00260-f004:**
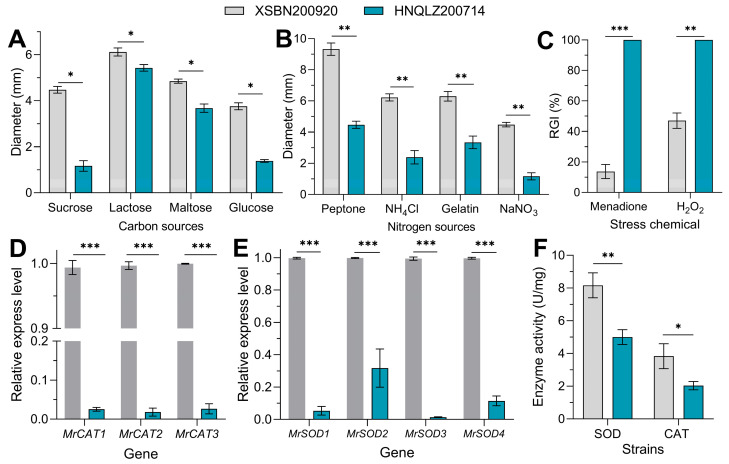
Growth and antioxidant stress levels of *Metarhizium rileyi* XSBN200920 and HNQLZ200214: (**A**) fungal colony diameters on different carbon sources; (**B**) colony diameters of fungi on different nitrogen sources; (**C**) the relative growth inhibition rates of menadione and H_2_O_2_ were RGI (%), respectively; (**D**) expression levels of *MrCAT* family genes in two strains of fungi; (**E**) *MrSOD* family gene expression levels of two fungi; (**F**) comparison of total CAT and SOD activity of two fungi strains; Tukey’s honestly significant difference [HSD]: *p* < 0.05 (*), *p* < 0.01 (**), *p* < 0.001 (***); error bars: standard deviation.

**Table 1 insects-14-00260-t001:** Primer sequences of real-time fluorescent quantitative PCR.

Primer Name	Primer Sequence (5′-3′)	Accession Number	Purpose
*MrCAT 1*	F:GAAGTTGACGATGGCGATGC R:TCACTTCTCGTCCTGCGCAA	OAA38085.1	RT-qPCR
*MrCAT 2*	F:CGCACACAACAACTTCTGGG R:GAAAGGCCTTCCAGTTCTCG	OAA43715.1	RT-qPCR
*MrCAT 3*	F:CGCACACAACAACTTCTGGG R:GTTCTTCGCGTTCTTCGTCG	OAA44228.1	RT-qPCR
*MrSOD 1*	F:CTTGGGTGGATGAAGGGCAT R:ACTCCTCGTTGTCACCCTTG	OAA47014.1	RT-qPCR
*MrSOD2*	F:AACGGCGAGATAGTATCCCA R:TCCTATACGCACTATCGACG	OAA42094.1	RT-qPCR
*MrSOD3*	F:TCGTGATGACTCATTGCAGC R:CGCATCCTCCAGCTTGATAT	OAA42645.1	RT-qPCR
*MrSOD 4*	F:GCTACATACGACTCCAGCCC R:TCAACGATCCGTCCTCGCAA	OAA40597.1	RT-qPCR
*ACTIN*	F:CTTTTAATCGGCGCACGGAG R:CGAAGCTTGGCGCTATTGTC	OAA40333.1	RT-qPCR (reference gene)

**Table 2 insects-14-00260-t002:** Toxicity of *M. rileyi* strain XSBN200920 and HNQLZ200714 against 3rd instar larvae of *S. frugiperda* larvae under different conidia concentrations (8 d).

Strain	Conidia Concentrations (Conidia/Ml)	Regression Equation	Correlation Coefficient (R Squared)
XSBN200920	1.0 × 10^4^	Y = 2.719X − 5.013	0.7479
	1.0 × 10^5^	Y = 5.207X − 8.824	0.7578
	1.0 × 10^6^	Y = 7.923X − 13.50	0.8455
	1.0 × 10^7^	Y = 13.64X − 21.49	0.9041
	1.0 × 10^8^	Y = 15.59X − 17.75	0.9499
HNQLZ200714	1.0 × 10^4^	Y = 1.506X − 2.704	0.7816
	1.0 × 10^5^	Y = 2.742X − 4.629	0.9247
	1.0 × 10^6^	Y = 4.038X − 5.752	0.9397
	1.0 × 10^7^	Y = 6.900X − 7.681	0.9737
	1.0 × 10^8^	Y = 8.395X − 8.214	0.9697

Note: Y means probit; X is time; “−” represents that the mortality of infected 3rd instar larvae was less than 50.00%, the same below.

**Table 3 insects-14-00260-t003:** Toxicity of *M. rileyi* strain XSBN200920 and HNQLZ200714 against 5th instar larvae of *S. frugiperda* larvae under different conidia concentrations (8 d).

Strain	Conidia Concentrations (Conidia/mL)	Regression Equation	Correlation Coefficient (R Squared)
XSBN200920	1.0 × 10^4^	Y = 1.739X − 2.725	0.8677
	1.0 × 10^5^	Y = 3.092X − 5.164	0.9155
	1.0 × 10^6^	Y = 5.664X − 10.28	0.9414
	1.0 × 10^7^	Y = 8.313X − 9.786	0.9704
	1.0 × 10^8^	Y = 9.554X − 9.090	0.9661
HNQLZ200714	1.0 × 10^4^	Y = 1.506X − 2.704	0.7816
	1.0 × 10^5^	Y = 2.742X − 4.629	0.9247
	1.0 × 10^6^	Y = 4.038X − 5.752	0.9397
	1.0 × 10^7^	Y = 6.900X − 7.681	0.9737
	1.0 × 10^8^	Y = 8.395X − 8.214	0.9697

Note: Y means probit; X is time; “−” represents that the mortality of infected 5th instar larvae was less than 50.00%, the same below.

## Data Availability

The data presented in this study are available in manuscript.

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
