# Peer review of "Virulence of Metarhizium rileyi Is Determined by Its Growth and Antioxidant Stress and the Protective and Detoxifying Enzymes of Spodoptera frugiperda"

_insects, 2023, doi:10.3390/insects14030260_

Round 1

Reviewer 1 Report

Pang et al sought to evaluate the virulence and biocontrol potential of two Metarhizium rileyi strains isolated from Spodoptera frugiperda. The results indicated two strain displayed difference in virulence against different developmental stages of the insect host. Also, the host response as well as fungal response was determined through biochemical and molecular analyses of protective enzymes and their genes.   The results indicated two strains induced different host responses and used different anti-oxidant genes. This study provides new understanding to the interaction between M. rileyi and S. frugiperda. This study is designed well and deserves publication in this journal.

Minor notes:

Line 2, 30. It is better to use ‘determined’ to replace ‘regulated’.

Line 12. Change ‘stages and instars’ into ‘developmental stages’.

Line 132. It is stated that this experiment is done at ‘different concentrations’. However, there is only one concentration of ‘1.0´10^8’. Please confirm this!

Line 157. Change ‘toxicity’ into ‘virulence’.

Line 301-302. This sentence is confusing. Two parts in this sentence has a little conflict. Please reword it!

Line 334. Change ‘demonstrated’ into ‘displayed’.

Line 381. It is better to use ‘mummified’ to replace ‘diseased’.

Line 434. It is suggested to add a conclusive remark at the end of discussion section.

Author Response

Thank you very much for your comments and suggestions. We have revised the parts with question in the manuscript. As the following:

Line 2, 30. It is better to use ‘determined’ to replace ‘regulated’.

Response: We have revised the manuscript.

Line 12. Change ‘stages and instars’ into ‘developmental stages’.

Response: We have revised the manuscript.

Line 132. It is stated that this experiment is done at ‘different concentrations’. However, there is only one concentration of ‘1.0´10^8’. Please confirm this!

Response: In our study, different concentrations of spore suspensions were used separately in various virulence assay experiments. For Morphological observation and colony growth rate measurement of Metarhizium rileyi,Spore suspensions with a concentration of 1.0×107 conidia /mL were used to inoculate fungi,Spore suspensions with a concentration of 1.0×106 conidia /mL were used to measure growth rate of fungi.

Line 157. Change ‘toxicity’ into ‘virulence’.

Response: We have revised the manuscript.

Line 301-302. This sentence is confusing. Two parts in this sentence has a little conflict. Please reword it!

Response: We have revised the manuscript.

Line 334. Change ‘demonstrated’ into ‘displayed’.

Response: Changed ‘demonstrated’ into ‘displayed’.

Line 381. It is better to use ‘mummified’ to replace ‘diseased’.

Response: We have revised the manuscript.

Line 434. It is suggested to add a conclusive remark at the end of discussion section.

Response: We have added the manuscript.

Reviewer 2 Report

The manuscript “Virulence is regulated by the growth and antioxidant stress of  the entomopathogenic fungus Metarhizium rileyi and the protective and detoxifying enzymes of Spodoptera frugiperda” is an interesting study with justified set of bioassays. However, I could not find the analysis details. I would encourage the authors to address and incorporate the following suggestions in the revised version to improve their manuscript.

·       Title of the study too long, better to shorten the title.

·       Line 16: “theoretical strategy” please be careful while using terminology

·       Line 18-19: Why “is one of the most destructive crop pests in the world” it is italicized, please correct it.

·       Line 19: “It poses a serious threat to world food production and security” Unnecessary please avoid repetition.

·       Abstract too long, please rewrite the abstract.

·       Line 72-74: Better to revise the statement for clarity “Following the invasion of the host body wall, the entomological fungi enter the blood cavity and also activate their superoxide dismutase (SOD) and catalase (CAT) in response to the environmental stress” because of ambiguity of activation of SOD and CAT due to environmental stress or pathogen attack.

·       Line 75-81: The use of future indefinite for an already explored mechanism is not an appropriate way in this context. I would appreciate it if the authors could revise the statement for clarity.

·       Line 87-88: The statement “This study, therefore, provides a new strategy to control Spodoptera frugiperda via biotechnology” needs to be revised to avoid unnecessary claims because the methods are being utilized for a long time.

·       Line 82-88: The authors should provide detailed objectives for clarity.

·       Line 93-100: Better to provide the year.

·       Line 107: Authors mentioned that they have used “striating methods” please cite it otherwise provide brief details of the method for clarity and broad readership.

·       Line 110-112: Please rewrite the sentence instead of just mentioning two media and their compositions, otherwise prepare the table as supplementary. However, the current expression is not appropriate.

·       Line 145: The words “eggs were” by mistake italicize. Please correct it.

·       Line 154-155: Better reword the heading for clarity.

·       Line 233: Table 1: Instead of Login number better to provide an accession number.

·       Heading 3.2 better to rewrite like a heading instead of a statement.

·       Section 2.9: The authors have used various methodologies and bioassays for this study. It would be better to explain in detail the assays/bioassays' statistical analysis details instead of a general statement that is ambiguous and not acceptable. Moreover, which software was used to analyze data?

·       Figure2 b-j: Why the authors did not apply lettering. * is not acceptable.

·       Figure 3: Please provide lettering

·       Figure 4: Please provide lettering

·       Discussion bit weak, and better to expand this sector by discussing each finding with regard to previous reports/studies.

·       The authors should provide a conclusion of the study for a broad readership.

·       Where the strains were deposited?

Author Response

Thank you very much for your comments and suggestions. We have revised the parts with question in the manuscript. See the attachment for details please.

Reviewer 3 Report

Very approximate knowledge of the International Code of Biological Nomenclature: this is the main problem of this paper, which needs to be revised also in the English frorm (see notes on attached file). For the rest, the topic is interesting and desipte the data look properly analyzed, the experimental design should be better detailed and explained

Author Response

Thank you very much for your comments and suggestions. We have revised the parts with question in the manuscript.

Round 2

Reviewer 3 Report

The authors seem to have taken on board the recommendations of the reviewers.